# Needs, Aggravation, and Degree of Burnout in Informal Caregivers of Patients with Chronic Cardiovascular Disease

**DOI:** 10.3390/ijerph17176427

**Published:** 2020-09-03

**Authors:** Elżbieta Szlenk-Czyczerska, Marika Guzek, Dorota Emilia Bielska, Anna Ławnik, Piotr Polański, Donata Kurpas

**Affiliations:** 1Department of Health Sciences, University of Opole, 68 Katowicka Street, 45-060 Opole, Poland; 2Medical and Diagnostic Centre (MDC), 08-100 Siedlce, Poland; marika.guzek@centrum.med.pl; 3Department of Family Medicine, Medical University of Białystok, 15-089 Białystok, Poland; d.bielska1@wp.pl; 4Faculty of Health Sciences and Social Sciences, Pope John Paul II State School in Biala Podlaska, 21-500 Biała Podlaska, Poland; lawnikania@gmail.com; 5Family physician’s practice, Non-Public Healthcare Center, 58-350 Mieroszow, Poland; p.polanski@wp.pl; 6Department of Family Medicine, Wrocław Medical University, 1 Syrokomli Street, 51-141 Wrocław, Poland; dkurpas@hotmail.com

**Keywords:** informal caregivers, met and unmet needs, growing burnout

## Abstract

This study aimed to answer three main questions with respect to home caregivers for people with cardiovascular disease: (1) Are the needs of home caregivers being met (and at what level)?; (2) what is the level of emotional exhaustion, depersonalization, and personal accomplishment of home caregivers?; (3) what sociodemographic variables of home caregivers are related to unmet needs and level of emotional exhaustion, depersonalization, and personal accomplishment? The study used the Camberwell Modified Needs Assessment questionnaire and the Maslach Burnout Inventory questionnaire. This study reports on 161 informal home caregivers of patients with cardiovascular disease. We found that younger caregivers were less likely to report unmet needs (*p* = 0.011), and showed lower rates of burnout on depersonalization and emotional exhaustion. In addition, caregivers who worked more often reported higher levels of met needs (*p* = 0.022), and showed lower rates of burnout on depersonalization (*p* = 0.005) and emotional exhaustion (*p* = 0.018). Subjects residing in urban areas were more likely to report unmet needs (*p* = 0.007), and showed higher rates of burnout on emotional exhaustion (*p* = 0.006). Older caregivers who are unemployed and reside in cities should be offered programs to determine their unmet needs and to receive support.

## 1. Introduction

The rising prevalence of cardiovascular diseases (CVD) has become a major health concern around the world. The impact of CVD should not only be considered from a medical perspective, but also be evaluated in terms of its social and economic impact. It appears that the most important factor in the increasing number of people being diagnosed with CVD is an aging population, as the incidence of these diseases increases significantly with age [1,2,3].

Health deterioration with age causes limitations in everyday life, the need for external support, and increased intensity of using health care services—these include care, treatment, and rehabilitation [4]. Therefore, it is reasonable to search for activities and support systems, both medical and environmental, which will help maintain optimal health and improve the quality of life of patients with chronic CVD. The analysis of burden, and increasing our knowledge about the biopsychosocial status of caregivers for people with CVD will make it possible to determine which caregivers are the most burdened, thus allowing practical support to be targeted towards specific groups of cares.

The care of chronically ill and/or elderly patients is a priority in most countries, and a patient’s home is considered to be the preferable location for giving care [5]. As the number of elderly and/or chronically ill individuals continues to grow globally, the role of ‘informal caregivers’ becomes more relevant. Informal caregivers are most often family members or friends who work for free and provide support to those in need [6].

In Europe, there are an estimated 100 million informal caregivers. The estimated care contribution of these caregivers exceeds the financial expenses of formal nursing care. In England and Wales alone, an estimated 950,000 residents over the age of 65 are thought to be informal caregivers, and 65% of these individuals (aged 60–94) experience long term health problems or disabilities themselves [7].

In Poland, it is clear that the majority of chronically ill and/or disabled individuals in need receive help from informal family caregivers [8]. According to the Central Office of Statistics, three out of four caregivers look after their own family members, are male, and reside in urban areas. In contrast, one-fifth of caregivers are females providing support to patients who are family members as well as individuals outside of their family [9].

Research clearly shows that informal caregivers play a substantial role in homecare. Despite increasing awareness of the key role of caregivers, there is a lack of knowledge on how to support them [10,11]. Caregivers often have more unmet needs than the patients they care for. Caregivers’ unmet needs affect their health and well-being, making it difficult for them to provide care in the way they would like [7]. Activities supporting caregivers of chronically ill and**/**or elderly patients should take into account the impact of the care provided on both the physical and mental condition of the caregivers. Proper reading of the caregivers’ needs will allow them to provide the necessary support, help increase their satisfaction with the role they play, and avoid care overload. As a result of exposure to chronic stress related with responsibility for the health and life, at the caregivers often develop depression, anxiety, sleep problems, report of headaches, spine problems, cardiac disease, and gastric ailments. The most common syndrome of burnout, i.e., psychophysical exhaustion, affects those caregivers who deal with an ill person alone [12].

Restrictions in both private and professional spheres, as well as health problems among caregivers (often due to a lack of system support) are reasons for the failure of models of care for chronically ill people, especially at home. Further, the support of elderly, chronically ill, and/or disabled informal caregivers requires more attention and recognition. There is a strong need for access to different forms of support services that caregivers might be able to choose from [13,14].

One previous study among caregivers of patients recovering from a cerebrovascular incident examined the impact of caregiver support on quality of life and perceived burden [14]. This study showed that, if not supported by health services, caregivers showed a deterioration in their own health. Further, females and subjects under 60 were most frequently exposed to stress. Previous research has also demonstrated that caregivers suffer more mental distress and the overload is more severe when they do not have an additional assistant, or when they have their own health issues [15]. Current recommendations for the homecare of elderly and/or chronically ill patients should be sustained; however, the needs, expectations, and health condition of caregivers should be taken into much greater consideration [10].

In Poland, the reality is that most chronically ill patients and those in the end-of-life phase remain at home and health care is provided by family members and other informal caregivers. However, there is a lack information regarding these informal forms of care [10]. In particular, there is a paucity of data in the literature regarding the relationship between sociodemographic factors and the needs and level of emotional exhaustion, depersonalization, and personal accomplishment in the care of CVD patients.

Considering the above, the purpose of this study was to analyze the level of needs and level of emotional exhaustion, depersonalization, and personal accomplishment in home caregivers for people with CVD, and to determine what sociodemographic variables of home caregivers are associated with these factors. Results of this study may help identify the individuals and families who need support in taking care of patients who are not self-reliant patients. Results may also inform the creation of evaluation tools to assess the quality of care and the needs and expectations of care from somatic, mental, social, and environmental perspectives.

## 2. Materials and Methods

### 2.1. Study Design

This study is a cross-sectional observational study. This study is part of a broader study to identify indicators that determine the effectiveness of home care for patients with chronic CVD, and to identify variables that determine effective support systems for their home caregivers. The study involved 350 patients with CVD. In order to define indicators specific to home care, 193 patients remained in the home under the care of primary care family nurses, while 157 patients went out to see their GP for follow-up visits. The study also included caregivers of patients under home care of primary care family nurses. The study involved 161 caregivers. This article presents a partial analysis of the results of this study on the level of needs and level of emotional exhaustion, depersonalization, and personal accomplishment in home care for people with CVD.

### 2.2. Sample

The study was conducted in Polish CVD patients and their informal caregivers. These patients received home care from a family nurse working in basic health care in the Opolskie, Dolnośląskie, Mazowieckie, Lubelskie, and Podlaskie provinces. Eight primary care institutions took part in the study. Patients and caregivers were encouraged to take part in the study by a family nurse during planned home visits. Respondents completed questionnaires in their home. Patients and caregivers were provided with one set of questionnaires each, and nurses completed an additional questionnaire concerning the patient (i.e., paired questionnaires with respect to the same patient). Data were collected from March 2016 to January 2017.

One hundred and eighty informal caregivers were invited to participate in the survey, but the final sample of participants was determined on the basis of their temporal availability. Finally, 161 people took part in the survey. The criteria for inclusion in the study were as follows: 18 years of age, and taking care of a patient with chronic CVD under home care at least 12 months before the study. The exclusion criteria (disqualification as determined by the family nurse) were cognitive disorders and other severe mental illness, and/or other difficulties preventing active participation in the study.

### 2.3. Variables and Data Collection

Needs of informal home caregivers of CVD patients were assessed with the modified version of the Camberwell Assessment of Need (CAN). The Camberwell Modified Short Needs Assessment questionnaire was developed by focus groups and competent judges and was applied to evaluate the needs of ED and general practice patients. The modification of CAN is focused on 22 problem fields [16]. In this research, the Camberwell Needs Index was calculated. The calculations consisted of the determination of the number (N) of met (1) and unmet (0) needs of the patient on the basis of 24 questions identifying 22 needs. Consecutively, within the number (N) of needs indicated by the studied person, the number (M) of met needs (1) was established. The M/N formula was used to calculate the Camberwell Index. In addition, the above method also makes it possible to calculate the Camberwell Index for unmet needs according to the formula: 1-M/N (not used in this analysis). The consistency for the modified version CAN questionnaire Cronbach’s alpha was 0.82 [16].

The present study also administered the Maslach Burnout Inventory (MBI) [17]. The MBI provides insight into three components of the burnout syndrome, which are divided into subscales, namely: (1) Emotional exhaustion (EE), (2) depersonalization (DE), and (3) personal accomplishment (PA). The MBI questionnaire contains 22 test items that are divided into the three burnout subscales, listed above. Nine of the 22 items correspond to the emotional exhaustion subscale, five items to the depersonalization subscale, and eight items correspond to personal accomplishment. The test items in emotional exhaustion and depersonalization subscales are formulated negatively, whereas items from the personal accomplishment subscale are positive. Each question is structured affirmatively and relates to the attitude and feelings that a participant may experience and rates on a 7-point scale (where 0 means never, and 6 means every day). The results are calculated separately for each subscale. A high level of burnout is characterized by high scores on the emotional exhaustion and depersonalization subscales and low scores on the personal accomplishment subscale [17]. According to Maslach and Jackson, the MBI can be used in professions that require contacting other people. It was, however, indicated, that after the adaptation, the questionnaire might be applied more broadly by changing some words, but the meaning of the scale remains the same [18].

To assess the sociodemographic aspects of the caregivers, an interview questionnaire was administered that assessed sex, age, marital status, education, place of residence, degree of relation to the patient, employment, and the period of care given. Of note, there were missing data from caregivers for the following variables: Age, marital status, family-related caregiver, employment, and period of caretaking. Thus, the numbers provided in the columns do not sum up to 161. Of note, for three cases, a non-family caregiver was both a neighbour and an informal partner.

### 2.4. Ethical Aspects

The study was approved by the Bioethical Commission at Medical University in Wroclaw (No. KB—86/2016).

### 2.5. Data Analysis

The results of the study were subject to statistical analysis with the use of R statistical package (version 3.4.0).

For the quantitative variables, the arithmetic mean, standard deviation, first quartile (Q.25%), median (Q.50%), third quartile (Q.75%), minimum, and maximum were calculated. For nominal variables, frequency (i.e., percent) was determined. The Shapiro-Wilk test showed that only two quantitative variables (i.e., age, personal accomplishment on the MBI) showed a normal distribution. The other included variables diverged from the normal distribution. The chi-square test was used to assess the qualitative variables.

The relationship between sociodemographic variables and the assessment of needs and level of emotional exhaustion, depersonalization, and personal accomplishment was analyzed using Spearman’s rank correlation coefficient, which does not require a normal distribution of the variables. The null hypothesis (H0) was tested wherein the Spearman’s rank correlation coefficient equaled 0. The alternative hypothesis was that the correlation coefficient differed from 0. The null hypothesis (H0) was rejected if the *p*-value was <0.05 (α = 0.05).

Next, for each explained variable, a separate logistic regression analysis, for at least nine different explanatory variables, was carried out to examine all possible models. For further analysis, only specific models that demonstrated significance were chosen. All variables in the model had to be statistically significant and included the largest number of explanatory variables, in the smallest number of models. Using the models selected, the odds ratio for the events examined were calculated and conclusions formulated on their basis. This approach did not require the use of model-matching procedures. The significance level was established at 0.05 (see Table 1).

## 3. Results

### 3.1. Sociodemographic Data of Caregivers

Most of the caregivers for CVD patients were women (70.2%; n = 113), and the median age was 55 years (min.–max.: 17–95; IQR: 42.50–55–64). The majority of caregivers were married (65.6%; n = 105), highly-educated (28.6%; n = 46 participants had a Master’s degree), or with secondary education (23.6%; n = 38), and resided in cities (58.4%; n = 94). Homecare was provided primarily by the patients’ parents (42.3%; n = 63), followed by spouses (28.2%; n = 42), and other relatives (18.1%; n = 27). In addition, 48.4% (n = 76) of caregivers were employed full-time, and 36.9% (n = 58) were currently unemployed. The median period of care given was four years (min.–max.: 1–51; IQR: 2–10) (see Table 2).

### 3.2. Assessment of the Level of Needs and Level of Emotional Exhaustion, Depersonalization, and Personal Accomplishment among Caregivers

The calculated Camberwell Index is 0.88 (min.–max.: 0.44–1). The median score on the emotional exhaustion subscale is 20 (min.–max.: 0–50); the median depersonalization score is 6 (min.–max.: 0–24), and the mean of personal accomplishment is 29.07 ± 8.43. These data suggest a moderate level of emotional exhaustion, a low level of depersonalization, and high level of personal accomplishment (see Table 3).

### 3.3. Significant Correlations

We tested for relationships between various sociodemographic variables and the assessment of needs. Needs met was negatively correlated with both age (r = −0.20; *p* = 0.011) and employment (r = −0.18; *p* = 0.022) such that a higher caregiver report of needs met was reported among younger and employed examinees. A lower rate of needs met was reported among caregivers residing in urban areas as compared to those in rural ones (r = 0.21; *p* = 0.007).

Next, we tested for associations between sociodemographic variables and level of emotional exhaustion, depersonalization, and personal accomplishment among caregivers. We found that level of emotional exhaustion, depersonalization, and personal accomplishment was positively associated with age, marital status, and employment. Level of emotional exhaustion, depersonalization, and personal accomplishment was negatively associated with education and place of residence. Lower results in emotional exhaustion and depersonalization subscales were more frequently found in younger (*p* = 0.010; *p* = 0.009), single or married (*p* = 0.02; *p* = 0.003) caregivers than in older, widowed or divorced caregivers. Caregivers residing in cities reported higher rates of emotional exhaustion (*p* = 0.006) than those living in the country. Higher values of depersonalization were observed among less educated (*p* = 0.028) caregivers as compared to well-educated caregivers. Employed caregivers showed lower rates of burnout on depersonalization (*p* = 0.005) and emotional exhaustion (*p* = 0.018) subscales than unemployed caregivers (see Table 4).

We also tested for a potential between burnout and unmet needs among caregivers. We found a significant relationship between unmet needs and burnout in all three MBI subscales (*p* < 0.001) (i.e., emotional exhaustion, depersonalization, personal accomplishment; see Table 5).

#### Method of Logistic Regression

The logistic regression performed in caregivers of CVD patients was also significant for unmet needs. Table 6 presents the results of the odds ratio in logistic regression model for the risk of unmet needs occurrence among caregivers. It was found that female caregivers are 2.51 times more likely to experience unmet needs than male caregivers.

Younger caregivers were 1.02 times less likely to report unmet needs than caregivers who were a year older, and were 6.07 times less likely than 78 year-old caregivers.

Caregivers with a rank in terms of marital status lower by 1 were 1.4 times less likely to report unmet needs than those with a higher category. In particular, single caregivers were 2.73 times less likely to report unmet needs as compared to divorced caregivers.

Caregivers with a higher rank in terms of residence were 1.4 times less likely to report unmet needs as compared to those with a residence rank lower by 1. In particular, rural residents were 2.77 times less likely to report unmet needs than those residing in big cities.

Caregivers with a higher rank in terms of education were 1.12 times more likely to report unmet needs than those with a lower educational rank. In particular, caregivers with primary education were 2.02 times less likely to report unmet needs than highly-educated caregivers.

Caregivers with a lower rank in terms of employment were 1.17 times less likely to report unmet needs than those in employment categories higher by 1. In particular, caregivers who were employed full-time were 2.24 times less likely to report unmet needs than unemployed caregivers.

The analysis of logistic regression in the group of caregivers of CVD patients was also significant for emotional exhaustion. Table 7 presents the results of the odds ratio in logistic regression model for the risk of emotional exhaustion occurrence among caregivers of CVD patients. Older caregivers were 1.02 times more likely to report emotional exhaustion occurrence than their younger counterparts, and were 3.32 times more likely to report emotional exhaustion than caregivers who were 78 years younger.

Caregivers with a lower rank in terms of residence were 1.37 more likely to report emotional exhaustion than those with a higher residence category. In particular, urban residents were 2.56 times more likely to report emotional exhaustion than those living in the countryside.

Caregivers who ranked lower in terms of education were 1.09 times more likely to report emotional exhaustion than those who ranked higher in terms of education. In particular, caregivers with primary education were 1.69 times more likely to report emotional exhaustion than highly educated caregivers.

Caregivers with a higher rank in terms of employment were 1.11 times more likely to report emotional exhaustion than those who ranked lower in the employment category. In particular, caregivers who were currently unemployed were 1.70 times more likely to report emotional exhaustion than full-time employees.

Next, we analyzed the logistic regression in the group of caregivers of CVD patients for depersonalization. Table 8 presents the results of the odds ratio in the logistic regression model for the risk of depersonalization occurrence among caregivers of CVD patients.

Caregivers who ranked lower in terms of residence were 1.21 times more likely to report depersonalization than those who ranked higher in the residence category. In particular, caregivers residing in cities were 1.77 times more likely to report depersonalization than those living in the countryside.

Caregivers who worked for a longer period of time were 1.04 times more likely to report depersonalization than those who provided care for a shorter period of time. Further, caregivers who provided 50 more years of care were 7.84 times more likely to report depersonalization than those who provided care for shorter periods.

Caregivers who ranked lower in terms of education were 1.14 times more likely to report depersonalization than those who ranked higher in the education category. In particular, caregivers with primary education were 2.15 times more likely to report depersonalization than highly educated caregivers.

Next, we evaluated results of the logistic regression in the group of caregivers of CVD patients for the personal accomplishment variable. Table 9 presents the results of the odds ratio in the logistic regression model for the risk of personal accomplishment occurrence among caregivers of CVD patients.

We found that females were 1.85 times more likely to report a decrease in personal accomplishment than their male counterparts.

Caregivers who ranked lower in terms of residence were 1.28 times more likely to report a decrease in personal accomplishment than those who ranked higher in this category. In particular, residents of big cities were 2.09 times more likely to report a decrease in personal accomplishment than those residing in the countryside.

## 4. Discussion

If caregivers have unmet needs, this affects their health and well-being, which can make it difficult for them to provide effective care in the way they want. The study showed, that half of the people in the surveyed population have at least 88% of their needs met. Care should be taken in interpreting the results due to the small size of the research group. Other researchers showed that 35% of support persons (i.e., informal caregivers) had at least one unmet need. Among this group, almost two-thirds reported many unmet needs (64.7%) [19]. Another study found that informal caregivers reported almost twice as many needs as the patients they care for [20]. Identification of needs enables problem identification and adequate intervention that can eliminate or reduce the need. The literature on the subject lacks research on the assessment of needs met (or unmet) by informal caregivers of people with CVD. Our results suggest the need for more research on unmet needs of caregivers of CVD patients to develop support services and adopt suitable interventions.

In our study, the level of emotional exhaustion, depersonalization, and personal accomplishment in caregivers showed to be at an average level. The highest results were found in the subscale that involved a sense of lack of personal accomplishments. Similar results were obtained when examining the burnout of family members in the care of a cancer patient covered by palliative home care [21]. There are other studies showing a high burnout rate among caregivers [18].

The current study indicates that older, unemployed caregivers who reside in urban areas were more likely to report unmet needs. We also demonstrated that female, older, divorced, well-educated, and unemployed caregivers who live in urban areas were more likely to report unmet needs. In the study, advanced age, marital status (i.e., divorced or widowed), and unemployment were related to higher reports of emotional exhaustion and depersonalization, whereas a lower level of education was associated with higher reports of depersonalization. The relationship between age and marital status and emotional exhaustion and depersonalization in caregivers is confirmed by other authors’ research [22]. In addition, depersonalization was more likely to be reported among older, unemployed, and less educated caregivers who reside in big cities. Higher depersonalization was reported among less educated city residents who have been providing care for a longer period of time. Personal accomplishment, in contrast, was more likely to be reported among female caregivers and those living in the countryside. Additionally, the study discovered a significant relationship between unmet needs and burnout, as measured by the MBI. This association was significant for all three MBI subscales (i.e., EE, DE, and PA). The chosen topic is still not very well described in the literature. It is difficult to find a comparison with any other research.

In our data, we found that 70.2% of all caregivers are women. This finding is similar to results of a prior study of caregivers of the elderly (74.8% women) [23], and in another study on caregivers from the chronically ill conducted by Marzec et al. (68.56% women) [24]. Together, these data suggest that there is a link between sex and caregiving. In most cases, we found that caregivers were over the age of 55, married, residing in urban areas, and were related to the patient (i.e., parents or spouses). It seems that among the caregivers of CVD patients younger people predominate, while our research has shown that the majority of care was provided by the patients’ parents— i.e., people older than patients. Young people are living faster and faster, so they care less about themselves, not realizing that the young age will not protect them from serious diseases. Among the caregivers there were people who indicated that they were adult children for their care recipients, however they constituted a small number of respondents. This requires further research, but at the same time it is a justification for the need to pay more attention to younger patients with CVD and their caregivers.

Research on environmental systems of social support and care for the elderly showed that informal caregivers most frequently report issues such as lack of time for care, and the need for institutional and personal support [23]. Further, prior research demonstrates that caregivers require information and training (e.g., first aid in emergency situations, practical advice on bedridden patient care, simple medical procedures, administering drugs, patting to prevent bedsores, providing basic massages and rehabilitation procedures, measuring blood pressure, among others), as well as, counselling and the availability of substitute care [7,10]. Another key issue is communication between health service workers and caregivers. Caregivers typically feel powerless and ignored if they fail to establish a relationship with medical professionals. This lack of coordination between health caregivers, as well as a lack of access to information has been clearly highlighted in studies and in consultation with interested parties [7,25].

The care of chronically ill patients causes long-term stress, and it can lead to exhaustion of the caregiver [26]. Caregiver burden is defined as the physical, emotional, material, and social costs incurred in caring for the chronically ill (Zarit, Bach-Reever, and Peterson, 1980) [27]. Burden and negative consequences in the area of health of caregivers negatively affect the quality of care, worsen the prognosis of patients, and increase the cost of care [28]. The results in this study provide the basis for more in-depth study and interest in these issues, and testify to the high demand for professional support (e.g., helpline, psychological counselling, support groups) to reduce the burden on caregivers.

It is important to note that it is the first study in Poland to report a relationship between sociodemographic factors and the needs and level of emotional exhaustion, depersonalization, and personal accomplishment among informal caregivers of CVD patients. In our study, we did not provide data on the health of patients and their needs. This may vary from case to case and affect the needs of caregivers. These correlations will be presented by the authors in another article. It seems, however, that these results are a good contribution to further research in this field, and identify people with specific characteristics that should be the target of programs to improve care for caregivers for people with CVD. The results presented here might be useful for creating systems of support for individuals and families who are seeking assistance in nursing patients who are not self-reliant. These data may also help inform the creation of care quality evaluation tools with the analysis of its needs and expectations in somatic, mental, social, and environmental aspects. Such examinations might improve the quality of life among caregivers, by helping them manage their own needs, as well as, their mental and physical wellbeing.

One limitation of the study is the relatively limited sample size, which limits our ability to generalize the results to the entire whole population of informal caregivers of CVD patients in Poland. This may be apparent in the relatively low values of the pseudo R2 indicator, which assesses the prediction of an explained variable with the use of a model. Here, the calculated pseudo R2 values ranged from 0.02 to 0.05 (0 ≤ pseudo R2 < 1), where the higher the value, the better the prediction. Nonetheless, results of the study are valuable and might be used in the course of interventions that support the development of a systemic model of home care for chronically ill patients. Future research should include a greater number of respondents and health care institutions.

## 5. Conclusions

Advanced age, sex, marital status, unemployment, place of residence, and high education of caregivers was related to more unmet needs among caregivers. Caregivers with the aforementioned characteristics should be targeted with programmes that aim to determine the category of unmet needs and provide tailor-made support in the relevant needs categories. Older, less educated, and unemployed caregivers who have been giving care for a longer period of time and reside in cities or in the countryside should be included in burnout prevention programmes. Fulfilling unmet needs in caregivers is important not only for planning supportive services, but also because there is a straightforward relationship between unmet needs and the exacerbation of emotional exhaustion, depersonalization, and personal accomplishment.

## Figures and Tables

**Table 1 ijerph-17-06427-t001:** Descriptive variables for logistic regression analysis models.

	Variables	Coding
x1	Gender	1—Women
		2—Men
x2	Age (in years)	Number of years
x3	Marital status	1—Single
		2—Married
		3—Widowed
		4—Divorced
x4	Place of residence	1—Big city <100 thousand inhabitants
		2—Middle town from 20–100 thousand inhabitants
		3—Town small >20 thousand inhabitants
		4—Village
x5	Education	1—Primary
		2—Vocational
		3—Secondary without Matura Exam
		4—Secondary with Matura Exam
		5—Post-secondary
		6—BA
		7—MA
x6	Family-related caregiver	1—Wife/husband
		2—Brother/sister
		3—Mother/father
		4—Uncle/aunt
		5—Cousin
		6—Other
x7	Non-family caregiver	1—Neighbour
		2—Informal partner
		3—Other
x8	Employment	1—Full time
		2—Part-time
		3—Sick leave-child care
		4 – Sick leave
		5—Unemployment benefit
		6—Unemployment
x9	Period of homecare	Number of years

**Table 2 ijerph-17-06427-t002:** Sociodemographic data of caregivers *.

Variable (n = 161)		n	%
Gender	women	113	70.2
	men	48	29.8
Age (in years)	n = 159		
	median	55	
	Q.25%–Q.50%–Q.75%	42.50–55–64	
	min.–max.	17–95	
Education	primary	16	9.9
	vocational	37	23
	secondary without Matura Exam	7	4.3
	secondary with Matura Exam	38	23.6
	post-secondary	11	6.8
	BA	6	3.7
	MA	46	28.6
	total	161	100
Place of residence	urban	94	58.4
	rural	67	41.6
	total	161	100
Marital status	single	31	19.4
	married	105	65.6
	widowed	15	9.4
	divorced	9	5.6
	total	160	100
Family-related caregiver	wife/husband	42	28.2
	brother/sister	6	4
	mother/father	63	42.3
	uncle/aunt	5	3.4
	cousin	6	4
	other	27	18.1
	total	149	100
Non-family caregiver	neighbour	9	60
	informal partner	4	26.7
	other	2	13.3
	total	15	100
Employment	full time	76	48.4
	part-time	10	6.4
	sick leave-child care	6	3.8
	sick leave	1	0.6
	unemployment benefit	6	3.8
	unemployed	58	36.9
	total	157	100
Period of homecare (in years)	n = 131		
	median	4	
	Q.25%–Q.50%–Q.75%	2–4–10	
	min.–max.	1–51	

Legend: n—group quantity; %—percentage; Q.25%—first quartile; Me—median; Q.75%—third quartile; Min.—minimum; Max.—maximum. * The figures in column n do not sum up to 161 due to missing data.

**Table 3 ijerph-17-06427-t003:** Assessment of the level of needs and burnout among caregivers.

Variable	n	M	SD	Q.25%	Me	Q.75%	Min.	Max.	*SW* Test *p*
Camberwell Index	161	0.83	0.13	0.76	0.88	0.94	0.44	1	<0.001
Emotional exhaustion	154 *	20.28	12.52	10	20	28	0	50	0.002
Depersonalization	155 *	7.22	6	2	6	11.50	0	24	<0.001
Personal Accomplishment	147 *	29.07	8.43	23	29	35	8	48	0.322

Legend: n—group quantity; M—mean; SD—standard deviation; Q.25%—first quartile; Me—median; Q.75%—third quartile; Min.—minimum; Max.—maximum; *p*—calculated level of significance for the standard Shapiro-Wilk test. * The figures in column n do not sum up to 161 due to missing data.

**Table 4 ijerph-17-06427-t004:** The relationship between variable sociodemographic data and the level of emotional exhaustion, depersonalization, and personal accomplishment among caregivers.

Variable	MBI Emotional Exhaustion		MBI Depersonalization		MBI Personal Accomplishment	
	*r*	*p*	*r*	*p*	*r*	*p*
Gender	−0.04	0.640	0	0.969	−0.04	0.647
Age (in years)	0.21	0.010	0.21	0.009	−0.08	0.336
Marital status	0.19	0.020	0.24	0.003	−0.09	0.303
Education	−0.11	0.164	−0.18	0.028	0.12	0.136
Family-related caregiver	−0.12	0.148	−0.06	0.492	−0.09	0.321
Non-family caregiver	−0.11	0.708	−0.08	0.775	−0.15	0.596
Employment	0.19	0.018	0.23	0.005	−0.09	0.264
Place of residence (urban/rural)	−0.22	0.006	−0.11	0.184	0.15	0.062

Legend: *p*—level of significance for test verifying null hypothesis, that r = 0 in contrary to r ≠ 0; r—Spearman correlation coefficient; r—if *p* ≤ 0.05.

**Table 5 ijerph-17-06427-t005:** Analysis between subscales Maslach Burnout Inventory (MBI) of the unmet needs.

Variable	Unmet Needs	
	*r*	*p*
MBI—Emotional Exhaustion	−0.47	*p* < 0.001
MBI—Depersonalization	−0.34	*p* < 0.001
MBI—Personal Accomplishment	0.32	*p* < 0.001

Legend: *p*—level of significance for test verifying null hypothesis, that r = 0 in contrast to r ≠ 0; r—Spearman correlation coefficient; r—if *p* ≤ 0.05.

**Table 6 ijerph-17-06427-t006:** Results of logistic regression analysis and odds ratio of logistic regression model in the group of caregivers. Explained variable: Unmet needs (0—more unmet needs, 1—fewer unmet needs) in caregivers.

Explanatory Variables		b_i_	SE*_i_*	z_i_	*p_i_* = Pr (> |z_i_|)			
Models with two explanatory variables								
**Model 1** (n = 161)								
Chi^2^ = 9.66, df = 2, *p* = 0.008, pseudo R^2^ =0.04								
	Intercept	–	–	–	–			
X1	Gender	0.919	0.320	2.874	0.004			
X2	Age	−0.023	0.008	−2.980	0.003			
**Model 2** (n = 161)								
Chi^2^ = 4.97, df = 2, *p* = 0.083, pseudo R^2^ = 0.02								
	Intercept	–	–	–	–			
X1	Gender	0.506	0.242	2.085	0.037			
X3	Marital status	−0.335	0.157	−2.134	0.033			
Models with three explanatory variables								
**Model 3** (n = 161)								
Chi^2^ = 11.15, df = 3, *p* = 0.011, pseudo R^2^ = 0.05								
	Intercept	–	–	–	–			
X4	Place of residence	0.340	0.115	2.955	0.003			
X5	Education	−0.117	0.059	−1.978	0.048			
X8	Employment	−0.161	0.063	−2.559	0.011			
**OR**		**Per Unit**			**Per Range**			
		OR	95% CI	1/OR	OR	95% CI	1/OR	range
**From Model 1**								
X1	Gender (1–2)	2.51	1.36–4.81	1.77	2.51	1.36–4.81	0.40	1
X2	Age (17–95)	0.98	0.96–0.99	0.99	0.16	0.05–0.52	6.07	78
**From Model 2**								
X3	Marital status (1–4)	0.72	0.52–0.98	1.40	0.37	0.14–0.90	2.73	3
**From Model 3**								
X4	Place of residence (1–4)	1.40	0.52–1.77	0.71	2.77	1.43–5.56	0.36	3
X5	Education (1–7)	0.89	0.52–0.99	1.12	0.50	0.24–0.98	2.02	6
X8	Employment (1–6)	0.85	0.52–0.96	1.17	0.45	0.24–0.82	2.24	5

Legend: OR—odds ratio; CI—95% confidence interval for OR. Chi-squared—statistical hypothesis test of Chi^2^ model adjustment; df—number of degrees of freedom; *p*—calculated level of test significance; pseudo R^2^—value which evaluates explanatory variable anticipation according to the model; b_i_—coefficient estimation in regression model; SE_i_—standard error estimation for b_i_ coefficient; z_i_—value of test statistics in standard distribution; (p_i_ = Pr (> |z_i_|)—calculated probability value p_i_ for double-sided critical area equal to z; n-group quantity.

**Table 7 ijerph-17-06427-t007:** Results of logistic regression analysis and the odds ratio in logistic regression model in caregivers. Explained variable: Emotional exhaustion (0—smaller, 1—bigger emotional exhaustion).

Explanatory Variables		b_i_	SE*_i_*	z_i_	*p_i_* = Pr (> |z_i_|)			
Models with two explanatory variables								
**Model 1** (n = 154)								
Chi^2^ = 8.51, df = 2, *p* = 0.014, pseudo *R^2^* = 0.04								
	Intercept	–	–	–	–			
X2	Age	0.015	0.006	2.607	0.009			
X4	Place of residence	−0.313	0.111	−2.818	0.005			
**Model 2** (n = 154)								
Chi^2^ = 5.29, df = 2, *p* = 0.071, pseudo *R^2^* = 0.03								
	Intercept	–	–	–	–			
X5	Education	−0.087	0.044	−1.996	0.046			
X8	Employment	0.106	0.052	2.045	0.041			
**OR**		**Per Unit**			**Per Range**			
		OR	95% CI	1/OR	OR	95% CI	1/OR	range
**from Model 1**								
X2	Age (17–95)	1.02	1.00–1.03	0.98	3.32	1.37–8.41	0.30	78
X4	Place of residence (1–4)	0.73	0.58–0.91	1.37	0.39	0.20–0.74	2.56	3
**from Model 2**								
X5	Education (1−7)	0.92	0.84–0.99	1.09	0.59	0.35–0.98	1.69	6
X8	Employment (1−6)	1.11	1.01–1.23	0.90	1.70	1.03–2.86	0.59	5

Legend: OR—odds ratio; CI—95% confidence interval for OR. Chi-squared—statistical hypothesis test of Chi^2^ model adjustment; df—number of degrees of freedom; *p*—calculated level of test significance; pseudo R^2^—value which evaluates explanatory variable anticipation according to the model; b_i_—coefficient estimation in regression model; SE_i_—standard error estimation for b_i_ coefficient; z_i_—value of test statistics in standard distribution; (p_i_ = Pr (> |z_i_|)—calculated probability value p_i_ for double-sided critical area equal to z; n-group quantity.

**Table 8 ijerph-17-06427-t008:** Results of logistic regression analysis and the odds ratio in logistic regression model in caregivers. Explained variable: Depersonalization (0—smaller, 1—bigger depersonalization).

Explanatory Variables		b_i_	SE*_i_*	z_i_	*p_i_* = Pr (> |z_i_|)			
Models with two explanatory variables								
**Model 1** (n = 155)								
Chi^2^ = 8.20, df = 2, *p* = 0.017, pseudo *R^2^* = 0.05								
	Intercept	–	–	–	–			
X4	Place of residence	−0.190	0.071	−2.664	0.008			
X9	Period of homecare	0.041	0.021	1.965	0.049			
**Model 2** (n = 155)								
Chi^2^ = 8.58, df =2, *p* = 0.014, pseudo *R^2^* = 0.05								
	Intercept	–	–	–	–			
X5	Education	−0.127	0.047	−2.701	0.007			
X9	Period of homecare	0.043	0.022	1.982	0.047			
**OR**		**Per Unit**			**Per Range**			
		OR	95% CI	1/OR	OR	95% CI	1/OR	range
**From Model 1**								
X4	Place of residence (1–4)	0.83	0.72–0.95	1.21	0.57	0.37–0.85	1.77	3
X9	Period of homecare (1–51)	1.04	1.00–1.09	0.96	7.84	1.19–79.55	0.13	50
**From Model 2**								
X5	Education (1–7)	0.88	0.80–0.96	1.14	0.47	0.26–0.80	2.15	6

Legend: OR—odds ratio; CI—95% confidence interval for OR. Chi-squared—statistical hypothesis test of Chi^2^ model adjustment; df—number of degrees of freedom; *p*—calculated level of test significance; pseudo R^2^—value which evaluates explanatory variable anticipation according to the model; b_i_—coefficient estimation in regression model; SE_i_—standard error estimation for b_i_ coefficient; z_i_—value of test statistics in standard distribution; (p_i_ = Pr (> |z_i_|)—calculated probability value p_i_ for double-sided critical area equal to z; n-group quantity.

**Table 9 ijerph-17-06427-t009:** Results of logistic regression analysis and the odds ratio in logistic regression model in caregivers. Explained variable: Personal accomplishment (0—smaller, 1—bigger personal accomplishment).

Explanatory Variables		b_i_	SE*_i_*	z_i_	*p_i_* = Pr (> |z_i_|)			
Models with two explanatory variables								
**Model 1** (n = 147)								
Chi^2^ = 6.05, df = 2, *p* = 0.049, pseudo *R^2^* = 0.03								
	Intercept	–	–	–	–			
X1	Gender	0.616	0.258	2.390	0.017			
X4	Place of residence	−0.246	0.116	−2.113	0.035			
**OR**		**Per Unit**			**Per Range**			
		OR	95% CI	1/OR	OR	95% CI	1/OR	range
**From Model 1**								
X1	Gender (1–2)	1.85	1.13–3.12	0.54	1.85	1.13–3.12	0.54	1
X4	Place of residence (1–4)	0.78	0.62–0.98	1.28	0.48	0.24–0.94	2.09	3

Legend: OR—odds ratio; CI—95% confidence interval for OR. Chi-squared—statistical hypothesis test of Chi^2^ model adjustment; df—number of degrees of freedom; *p*—calculated level of test significance; pseudo R^2^—value which evaluates explanatory variable anticipation according to the model; b_i_—coefficient estimation in regression model; SE_i_—standard error estimation for b_i_ coefficient; z_i_—value of test statistics in standard distribution; (p_i_ = Pr (> |z_i_|)—calculated probability value p_i_ for double-sided critical area equal to z; n-group quantity.

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
