# Peer review of "Needs, Aggravation, and Degree of Burnout in Informal Caregivers of Patients with Chronic Cardiovascular Disease"

_ijerph, 2020, doi:10.3390/ijerph17176427_

Round 1

Reviewer 1 Report

Thank you very much for the opportunity to review this manuscript! Overall, there has been a substantial improvement in the quality of writing compared to the first submission, the authors have done a wonderful job with enhancing the quality of this manuscript!

Below are my comments:

1) Abstract: no “and” between questions – please put a semicolon instead

2) L 48: “health” before “caregivers” should be removed

3) L 62-63, please re-phrase as “on-fifth of carers are females providing support to patients who are family members, as well as individuals outside of their family.”

4) This is minor, but it is best to adhere to one term throughout the paper – either “caregiver” or “carer”

5) L73-75, please correct punctuation and delete unnecessary articles “the”: “As a result of exposure to chronic stress related to the responsibility for the health and life of the charge, carers often develop depression, anxiety,  sleep problems, headaches, spine problems, cardiac disease, and gastric ailments.” It is best not to say “complain” but “report” instead.

6) L 71-72: may omit this sentence

7) L 103: no need for comma before “from”

8) L 123: “data” are plural, so please re-state as “data were collected . . .”

9) L 285: please replace “are employed” with “were employed”

10) Please edit the whole text to decrease word count wherever possible and delete repetitions, you can avoid repeatedly stating “the present study.”

11) Conclusion, L 419: unless I am mistaken, I understood that lower level of education was associated with more unmet needs – not high education?

12) L 365-366: no need to report p-value here

13) L 368-369: please delete this sentence, it is obvious that larger sample sizes are always preferable

14) L 370-371: I would delete this sentence as the literature on caregiver burden is abundant, and it is unclear at least to me how caregiver burden is different from caregiver burnout syndrome

15) L 377-379: this is too short of a paragraph – please add additional sentences or combine this paragraph with others. Overall, I found this surprising and almost paradoxical – it is rare to find reports where caregivers of persons who have diseases associated with aging (e.g., CVD) are parents of those patients – not adult children

16) Did you have any caregivers who were adult children for their care recipients? There is no mention of this in the table, but overall this is very surprising – it is perfectly valid to report and it would be interesting to see more discussion about why parents care for their children with CVD – seems quite unusual and a discussion would be interesting

17) L 394-395: “worsen patients” – unclear

18) Although you provide a lot of tables, I would suggest that you incorporate another table or figure that summarizes all findings in a succinct manner (e.g., bullet points) – something that the reader can easily take away, as the findings appear a bit disparate -many characteristics are associated with the dependent variables that were examined

Author Response

Dear Reviewer,

Additionaly, we would like to sincerely thank the Editorial Board and the Reviewer of your esteemed International Journal of Environmental Research and Public Health for their positive feedback and constructive recommendations improving our observational paper entitled Needs, Aggravation, and Degree of Burnout in Informal Care Providers of Patients with Chronic Cardiovascular Disease by Elżbieta Szlenk-Czyczerska, Marika Guzek, Dorota Emilia Bielska, Anna Ławnik, Piotr Polański and Donata Kurpas.

In this second round of revision, we focused our efforts strongly on the points made in your letter. We would like to respond to this opinion based on our careful revision, point by point, as you can see in the table below. Accordingly, the final version of the manuscript text includes the all necessary modifications and improvements.

Kind regards,

Elżbieta Szlenk-Czyczerska

Reviewer 2 Report

I still think that the effects are small that the study suffers from data regarding how statsu of the patients and the needs of the patients. This might differe between diffrent cases and affect needs of th acare giver 

Author Response

Dear Reviewer,

Additionaly, we would like to sincerely thank the Editorial Board and the Reviewer of your esteemed International Journal of Environmental Research and Public Health for their positive feedback and constructive recommendations improving our observational paper entitled Needs, Aggravation, and Degree of Burnout in Informal Care Providers of Patients with Chronic Cardiovascular Disease by Elżbieta Szlenk-Czyczerska, Marika Guzek, Dorota Emilia Bielska, Anna Ławnik, Piotr Polański and Donata Kurpas.

In this second round of revision, we focused our efforts strongly on the points made in your letter. We would like to respond to this opinion based on our careful revision, point by point, as you can see in the table below. Accordingly, the final version of the manuscript text includes the all necessary modifications and improvements.

Elżbieta Szlenk-Czyczerska

Reviewer 3 Report

Dear authors,

Thank you for addressing my comments and doubts.

Kind regards

Author Response

Dear Reviewer,

the authors thank the reviewer for a positive feedback and constructive recommendation improving the quality of our paper. 

Kind regards,

Elżbieta Szlenk-Czyczerska

This manuscript is a resubmission of an earlier submission. The following is a list of the peer review reports and author responses from that submission.

Round 1

Reviewer 1 Report

Thank you for the opportunity to review this paper. The authors have done a thorough job with explaining the study and providing much detail. Below are my comments:

1) It may be best to spell out “cardiovascular disease” in the abstract before using an abbreviation

2) The first question is really 2 questions (whether the needs are met and what is the severity of burnout) – may be best to separate the questions

3) The standard abbreviation is CVD, not CVDs

4) It may be best to commit to using one term throughout the paper, such as “carer” or “caregiver” but not mix “carer” with “care provider”

5) Last sentence in the abstract needs to be re-phrased: “Older caregivers who are unemployed and reside in cities should be offered programs to determine their unmet needs and to receive support” (“personalized” may not be a necessary adjective, since it is presumed that support is personalized)

6) Page 2, lines 60-61: you could rephrase these sentences as “One-fifth of caregivers are females and those who reside in . . . “ Current phrasing makes the reader re-read these sentences.

7) Throughout the paper: rather than stating “home environment,” you may simply state “home”

8) Page 3, line 117: no need to specify in parentheses “disqualification …” as exclusion criteria is an established term

9) Page 3, line 118: “intensified” mental disorders does not seem to be an accepted term, do you mean severe mental illness?

10) CANSAS – please spell out the first time before using the abbreviation

11) What is the Camberwell index and why was it used? Perhaps, it could be omitted, as not all readers may know this index, it is unclear in the formula what is M and N

12) Page 3, line 131, no need to state “i.e., Cronbach’s alpha” – simply state Cronbach’s alpha was 0.82.

13)  Page 3, line 133: no need to state “which was developed by . . .”, simply cite the reference

14) Page 4, line 151: you already stated on page 3 that 161 informal caregivers was the final sample size. Also, sentences should not begin with a number, if a number is used, it must be spelled out

15) Lines 156-159: it is generally accepted that participation in studies is voluntary and that participants are informed about it, so you may delete the last 2 sentences, only leaving the sentence about the ethical committee approval

16) Line 163: second quartile is the median, so it is best to use the term “median” as it is used more commonly rather than “2nd quartile”

17) Line 164: accepted term is “nominal” or “categorical” variable rather than “qualitative variable”

18) Unless required by the journal, it is unnecessary to include 2 zeros after the decimal in tables and in the Results

19) In the description of demographics, when you state that caregivers were highly educated, you need to specify in the parentheses that 28.6% participants had a Master’s degree, or you may want to combine BA and MA recipients, otherwise, it is unclear from the text what 28.6% signifies

20) I find it interesting that most care was provided by patients’ parents – that is parents who are even older than patients? I believed patients in this sample are older adults with cardiovascular disease, so appears unusual that their parents are taking care of their older adult children

21) Table 2: period in homecare – standard deviation is greater than the mean, if you subtract 9.22 from 7.20 you get a negative number – this is impossible given that period of homecare cannot have negative values. This distribution is likely highly positively skewed. Skewed data must be described with median, IQR, minimum, and maximum – not with Mean and SD. Also, no need to repeat abbreviations both in the table and in the legend, customarily, they are provided only in the legend

22) By cutting down on words elsewhere in the paper, you may save room for words here: it may be better not to abbreviate DE, PA, and EE – it is hard to read with so many abbreviations, best to keep it spelled out throughout the paper

23) The use of the term “decreased level of personal accomplishment” is confusing – it is best to simply state “personal accomplishment” -  how do you know if it is decreased unless there were 2 measurements and a difference was calculated

24) Unclear what p<0.001 signifies on line 193

25) To conserve space, you may delete Table 4, simply reporting significant correlations

26) Line 219 missing “with” age …

27) It is customary to begin the discussion section with addressing your research questions (needs, severity of burnout, and sociodemographic variables related to unmet needs and severity of burnout) – do not begin discussion with the description of your sample

28) When you discuss burden (lines 401-408), although indeed the term burden has several variations, you may cite Zarit, Bach-Reever, & Peterson (1980) as this is the landmark paper establishing caregiver burden. Zarit and colleagues do provide definition of burden.

29) Line 428: decreased level of needs met – hard to read, may restate as “more unmet needs”

30) Too many tables – if at all possible, please decrease the number of tables and decrease the text in legend

Overall, throughout the paper: there are numerous instances where text could be paraphrased to shorten it – state the same idea with fewer words.

Reviewer 2 Report

The subject matter is of interest. 

Results may also inform the creation of evaluation tools to assess the quality of care, and the needs and expectations of care, from somatic, mental, social, and environmental perspectives."

The data collected suffer from that important information is not given. For example  What is the status of the patients i.e do what are the needs of the patients?

They produce a lot of regressions most of the correlations are low with the possible excepton of those presented in table 6. The model analysis in tables 7, 8, 9, 10 gives results with coefficient of determantion of max 0.05.

My impression is that these values are not sufficiently strong to buld a model on.

The mansucript is also rather long and could be condensed just giving the most central results.

Reviewer 3 Report

Dear authors,

The study is interesting and easy to read and understand. However, I have some questions and comments that should be solver.

  • You talk all the time in the manuscript about "severity of burnout" but the MBI does not measure the severity of burnout. It just measure if your EE, D or PA is high, medium or low so please modify the word "severity" and use the word "level of EE, D or PA".
  • The introduction should include a bit of information about needs and burnout.
  • Please do not use acronyms in the title and the abstract.
  • The material and methods section should be divided into subsections to facilitate the reading (design, sample, variables and data collection, ethical aspects, data analysis).
  • You say "The shapiro-wilk test showed that only two qualitative variables" but age and PA are not qualitative variables.
  • Table 1 must be cited in the text.
  • In the line 190 you say "homecare was provided primarily by the patients´parents". Is that correct? Usually, parents are cared by their sons.
  • Is there any possibility to combine the Odds ratio tables? There are too many tables for 1 manuscript.
  • I do not understand where the p values from table 3 come.
  • In line 231 you say "main results". I think that the other results are also main results.
  • The second paragraph of the discussion should be the first paragraph of the discussion.
  • I think that the discussion repeat a lot of the information about the  results and does not discuss all of them.

Kind regards